# Adverse Prognostic Impact of Diagnostic Ureterorenoscopy in a Subset of Patients with High-Risk Upper Tract Urothelial Carcinoma Treated with Radical Nephroureterectomy

**DOI:** 10.3390/cancers14163962

**Published:** 2022-08-17

**Authors:** Ichiro Yonese, Masaya Ito, Yuma Waseda, Shuichiro Kobayashi, Masahiro Toide, Ryoji Takazawa, Fumitaka Koga

**Affiliations:** 1Department of Urology, Tokyo Metropolitan Cancer and Infectious Diseases Center Komagome Hospital, Tokyo 113-8677, Japan; 2Department of Urology, Tokyo Metropolitan Ohtuska Hospital, Tokyo 170-8476, Japan

**Keywords:** upper tract urothelial carcinoma, ureterorenoscopy, radical nephroureterectomy, oncological outcomes

## Abstract

**Simple Summary:**

Although adverse oncological effects of ureterorenoscopy (URS) on disease progression have been of concern, no study has demonstrated such effects in patients with upper tract urothelial carcinoma (UTUC) undergoing radical nephroureterectomy (RNU). The present retrospective study included 143 UTUC patients between 2010 and 2021 at two tertiary care hospitals, of whom 79 received URS prior to RNU. Subgroups were stratified by clinicopathological variables relevant to prognosis. No significant prognostic difference was found between patients with and without URS in the entire cohort. Subgroup analysis demonstrated that URS was significantly associated with worse overall (*p* < 0.001) and progression-free survival (*p* = 0.008) in patients with non-papillary and ≥pT3 UTUC. URS was rather associated with better PFS in those with papillary and ≤pT2 UTUC. Diagnostic URS may better be avoided in patients with high-risk UTUC features unless URS is necessary for diagnosing UTUC.

**Abstract:**

Background: We hypothesized that diagnostic ureterorenoscopy (URS) may adversely affect prognosis in a subset of patients with high-risk upper tract urothelial carcinoma (UTUC) undergoing radical nephroureterectomy (RNU). Methods: The present retrospective study included 143 patients with UTUC treated between 2010 and 2021 at two tertiary care hospitals, of whom 79 received URS prior to RNU. Subgroups were stratified by clinicopathological variables relevant to prognosis. The primary endpoint was to evaluate the prognostic impact of URS on overall survival (OS) and progression-free survival (PFS) after RNU. Results: During follow-up (median 54 months for survivors), 32 cases of all-cause mortality and 40 cases of progression were recorded. No significant difference was found in OS or PFS between patients with and without URS. Subgroup analysis demonstrated that URS was significantly associated with worse OS (*p* < 0.001) and PFS (*p* = 0.008) in 29 patients with non-papillary and ≥pT3 UTUC. Importantly, URS did not have any adverse effects on prognosis in 62 patients with papillary and ≤pT2 UTUC (*p* = 0.005). Conclusions: URS may adversely affect prognosis of UTUC patients, specifically non-papillary and ≥pT3 disease. URS may better be avoided in patients with high-risk UTUC features unless URS is necessary to diagnose UTUC. This study also corroborates the oncological safety of URS in those with low-risk UTUC.

## 1. Introduction

Upper tract urothelial carcinoma (UTUC) is uncommon, having an estimated annual incidence of only two per 100,000 population in the West and accounting for 5–10% of all cases of urothelial carcinoma [1]. Surgery is the standard of care for patients with non-metastatic UTUC. The recent EAU guidelines recommend kidney-sparing therapy for patients with low-risk disease (unifocal, non-invasive, low-grade disease <2 cm) or those with serious renal insufficiency or a solitary kidney [1]. Otherwise, radical nephroureterectomy (RNU) remains the mainstay of surgical treatment.

Diagnostic ureterorenoscopy (URS) is useful for determining the presence, appearance, and size of UTUCs, especially in cases where the tumor is not visible on imaging studies or where urine cytology returns negative [2]. A URS biopsy can also determine the tumor grade in more than 90% of cases [3]. Combining the URS biopsy grade, imaging findings, and urine cytology can be helpful in selecting candidates for kidney-sparing therapy. The survival rates in patients with low-risk UTUC receiving kidney-sparing therapy and RNU are comparable [4].

Despite its diagnostic utility, URS is known to increase intravesical recurrence after RNU [2,5,6]. Although previous meta-analyses demonstrated no adverse prognostic impact of URS prior to RNU on survival or progression [2,7], some studies have raised concerns about lymphovascular dissemination of tumor cells via increased intra-ureteropelvic pressure during URS [8,9]. We hypothesized that diagnostic URS may adversely affect the prognosis in a subset of patients with biologically aggressive, high-risk UTUC. To the best of our knowledge, no previous study has investigated the details of the prognostic impact of URS in subgroups stratified by clinicopathological variables relevant to the biological aggressiveness of the disease. The present study therefore retrospectively investigated the prognostic impact of diagnostic URS in non-metastatic UTUC patients undergoing RNU.

## 2. Materials and Methods

### 2.1. Study Design and Data Collection

The present retrospective, bicentric study enrolled patients with UTUC treated with RNU between January 2010 and March 2021 at Tokyo Metropolitan Cancer and Infectious Diseases Center at Komagome Hospital and Tokyo Metropolitan Ohtsuka Hospital. Patients who underwent multiple URS with kidney-sparing therapy before their RNU, those who received neoadjuvant therapy, and those with a follow-up period <6 months were excluded. The study protocol was approved by the Institutional Ethical Committee at the two participating hospitals.

URS was strongly recommended to all of patients with negative VUC or equivocal imaging findings to assess the presence or absence of UTUC and to localize the disease. Otherwise, URS was offered as an option to confirm UTUC presence endoscopically and histologically. Following retrograde urography, a semirigid ureterorenoscope (6/7.5 Fr semirigid, Wolf, Knittlingen, Germany) or flexible ureterorenoscope (8.3/9.9 Fr URF-V, 8.5/8.4 Fr URF-V2, Olympus, Tokyo or 7.5/8.4 Fr Flex-X2, Storz, Tuttlingen, Germany) was transurethrally inserted into the target lesions, and the tumors were biopsied whenever possible. RNU was performed via open or minimally invasive surgery (April 2013 or later at Komagome Hospital). Lymph node dissection was considered in patients with ≥cT3 disease or suspicion of nodal involvement.

The clinical variables included age, sex, ECOG-PS, BMI, smoking history, bladder cancer history, tumor location, clinical T and clinical N stage, and tumor size based on preoperative imaging studies; the presence or absence of hydronephrosis; voided urine cytology (VUC); whether or not a URS biopsy was performed; surgical modality employed in the RNU; and adjuvant chemotherapy. VUC was analyzed according to the Paris System [10] and the Papanicolaou classification system at Komagome Hospital and Ohtsuka Hospital, respectively. High-grade UC, low-grade urothelial neoplasm, suspicion of high-grade UC for the former, and class V and IV for the latter were considered to indicate positivity. The pathological variables included tumor multifocality, macroscopic tumor configuration (papillary or non-papillary), pT and pN stage, histological grade, lymphovascular invasion, soft-tissue surgical margin status, and the presence or absence of a variant histology. The tumors were staged according to the 2002 TNM system and graded according to both the 1973 WHO classification and the 2004 WHO/International Society of Urological Pathology classification at Komagome Hospital and the 1973 WHO classification at Ohtsuka Hospital. Thus, the 1973 WHO grading system was used in this study. Platinum-based, systemic adjuvant chemotherapy was considered for patients with ≥pT3 or pN+ disease.

### 2.2. Outcomes

The primary endpoint was the assessment of the impact of URS on OS and PFS after RNU. Progression was defined as a local recurrence and metastases to the lymph nodes and distant organs after RNU. Intravesical recurrence was excluded from progression.

### 2.3. Statistical Analysis

Differences in categorical and continuous variables were evaluated using Fisher’s exact test and Wilcoxon rank-sum test, respectively. OS and PFS were calculated from the date of RNU to all-cause mortality and progression, respectively, or the last follow-up. Survival curves were drawn using the Kaplan–Meier method and compared using the log-rank test. The Cox proportional hazards model was used to assess for a potential association of the variables with survival. A reduced multivariable model was developed using the stepwise backward method. All statistical analyses were performed using R (version 4.0.5, R foundation For Statistical Computing, Vienna, Austria) and EZR (version 1.54) [11]. Two-tailed *p* < 0.05 was considered to indicate statistical significance.

## 3. Results

### 3.1. Baseline Characteristics

The present study included 143 patients (82 at Komagome Hospital and 61 at Ohtsuka Hospital), as listed in Table 1. All the patients had UTUC, and 13 (9.1%) had a variant histology. Sixty-five (45.5%) and six (4.2%) patients had ≥pT3 and pN1, respectively; of these, 25 (17.5%) received platinum-based adjuvant chemotherapy.

Of the 143 patients, 79 (55.2%) underwent URS prior to RNU. The median interval (IQR) between URS and RNU was 5.4 (3.7–10) weeks. Proportions of patients undergoing URS were significantly higher in male (*p* = 0.036), ≤cT2 (*p* = 0.046) and cN0 disease (*p* = 0.038), and smaller tumor size (*p* < 0.001). No significant difference was observed in the pathological variables between patients with and without URS (Table 1).

### 3.2. Survival Analyses in the Entire Cohort

During the follow-up period (median 54 months for survivors; IQR: 24–68 months), 32 cases of all-cause mortality and 40 cases of progression were observed. The median times (IQR) to all-cause mortality events and progression events were 25 months (17–34 months) and 10 months (7–14 months), respectively. The progression cases consisted of five (3.5%) local recurrences, 23 (10.5%) lymph node metastases, and 20 (14.0%) visceral metastases (18 lung, 5 liver, and 2 bone); of these, the lymph node and visceral metastases concurrently developed in eight patients. On multivariable analysis, ≥pT3 was independently associated with shorter OS (HR: 5.24; *p* < 0.001) and PFS (HR: 3.55; *p* = 0.005), and lymphovascular invasion was independently associated with shorter PFS (HR: 3.10; *p* = 0.004; Table 2). No significant difference in OS or PFS was observed between patients with and without URS.

### 3.3. Subgroup Survival Analyses

Next, the prognostic effects of URS were examined in each clinicopathological subgroup. Figure 1 shows that URS was significantly associated with shorter OS and PFS in subgroups with ≥cT3 disease (univariable HR and 95% CI: 3.87 [1.20–12.49] and 5.74 [2.05–16.03], respectively), non-papillary configuration (univariable HR and 95% CI: 6.57 [1.46–29.43] and 3.93 [1.42–10.88], respectively), and ≥pT3 (univariable HR and 95% CI: 2.79 [1.16–6.68] and 2.55 [1.20–5.39], respectively). The clinical T stage on imaging studies was inaccurate, with an understaging rate of 55% for ≥pT3 disease (Table 3). Hence, clinical T stage was excluded from the subsequent subanalyses.

The entire cohort was divided into four groups by the presence or absence of non-papillary configuration and ≥pT3 (Figure 2A). URS was significantly associated with worse OS (*p* < 0.001) and PFS (*p* = 0.008) in the subgroup with non-papillary and ≥pT3 UTUC (n = 29, Figure 2B). No significant adverse prognostic effect of URS was observed in the subgroup with papillary and ≥pT3 UTUC (n = 36, Figure 2C) or non-papillary and ≤pT2 UTUC (n = 16, Figure 2D). Importantly, URS did not show any adverse prognostic effects but was rather associated with significantly better PFS (*p* = 0.005) in the subgroup with papillary and ≤pT2 UTUC (n = 62, Figure 2E).

In the subgroup with non-papillary and ≥pT3 UTUC, the proportion of hydronephrosis cases was significantly higher (*p* = 0.041), and the pack year was significantly lower (*p* = 0.035), in patients with URS than in those without URS (Table 4). During the follow-up period, 11 cases of all-cause mortality and 16 cases of progression were observed in this subgroup. Of these, there were 13 cases of visceral or lymph node metastasis. An adverse prognostic effect of URS on PFS was still observed after adjusting for hydronephrosis (adjusted HR: 3.83; *p* = 0.041; Table 5).

## 4. Discussion

The present bicentric, retrospective study, in which more than half the patients received diagnostic URS prior to RNU, demonstrated that URS was significantly associated with worse OS and PFS in the subgroup with non-papillary and ≥pT3 UTUC. Importantly, such adverse oncological effects were not observed in patients with papillary and organ-confined (≤pT2) UTUC. These findings corroborate the oncological safety of URS for low-risk UTUCs in terms of survival and progression. However, diagnostic URS may have to be avoided as much as possible in cases where non-papillary and ≥pT3 UTUC is probable.

Previous studies, including meta-analyses, have claimed increased intravesical recurrences after diagnostic URS before RNU [2,5,6,7,12,13,14,15], as is well-known in the urological community. However, although the potential for adverse oncological effects of URS on disease progression has attracted attention and has been hotly debated [9], cumulative studies have not demonstrated any significant association [2,5,7,13,14,15,16,17,18,19,20]. However, these studies did not perform detailed subgroup analysis. The present study demonstrated a significant association between diagnostic URS and poorer OS and PFS in a subgroup of patients with non-papillary and ≥pT3 UTUC, whereas the prognostic significance was not demonstrable for the entire patient cohort. These findings suggested that adverse oncological effects of URS are specific to the pathological phenotypes of UTUC.

Some studies have cautioned against disseminating tumor cells by increasing intra-ureteropelvic pressure during URS [8,9]. Our findings suggested that this pathophysiological process may depend on the pathological phenotype. In contrast to papillary, non-invasive UC, non-papillary, invasive UC has a much greater potential for epithelial to mesenchymal transition, one of the initial, biological processes in systemic tumor dissemination [21]. The intratumoral density of microvessels and lymphatic vessels is also higher in more invasive forms of UTUC [22,23]. Thus, tumor cells are prone to invading the lymphovascular circulation in non-papillary and ≥pT3 UTUC. In fact, in the present study lymphovascular invasion in RNU specimens was detected more frequently in non-papillary and ≥pT3 disease (59%, 17/29) than in papillary or organ-confined disease (28%, 32/114; *p* = 0.004). Moreover, owing to their architectural properties, tumoral microvessels and lymphatic vessels may be more vulnerable to penetration by tumor cells via increased intra-ureteropelvic pressure in non-papillary, than in papillary, UTUC.

The current EAU guidelines recommend kidney-sparing therapy for patients with low-risk UTUC [1]. In the present study, diagnostic URS was unlikely to compromise the oncological outcomes in terms of OS and PFS in a subgroup of papillary and organ-confined UTUC. This finding corroborates the oncological safety of URS in patients with low-risk UTUC who almost invariably need URS for evaluating the appearance, multifocality, and histological grade.

Inaccuracies in image-based, clinical T staging remain a major problem in the current management of UTUC. In the present study, 55% of cases of ≥pT3 disease were understaged on imaging studies. Given the possible, adverse, oncological effects of diagnostic URS on non-papillary and ≥pT3 UTUC, it may be oncologically safe to avoid URS in patients with any high-risk UTUC features, including high-grade cytology and hydronephrosis, even if the image-based clinical T stage is ≤T2. URS should be considered only when a diagnosis of UTUC cannot be made with urine cytology or imaging studies including CT or MRI urography [1].

Although the present study was the first to demonstrate adverse oncological effects of URS in a subset of patients with aggressive UTUC, it has some limitations. First, our findings were derived from a retrospective, bicentric study enrolling a relatively small patient cohort. Second, there was possible selection bias in association with indications of URS. Although male patients and those with ≤cT2 and cN0 disease received URS more frequently in the present study, these features were unlikely to be associated with unfavorable prognosis [24]. Third, the present study lacked a central pathology review; tumor configuration and the presence of ≥pT3 disease were the key pathological features relevant to assessing the prognostic effects of URS. According to a previous study investigating the impact of central pathology review in a multicentric cohort of UTUC patients, review and local pathologists did not differ in their assessment of the pathological demographics of tumor configuration and ≥pT3 disease [25]. The same study also demonstrated no prognostic difference between patients with ≥pT3 disease recorded by the review pathologist and those recorded by local pathologists. Thus, the lack of central pathology review may not have substantially influenced the results of our study. Forth, our findings cannot be used for predicting patients in whom URS may worsen oncological outcomes after RNU. Despite these limitations, we believe that the findings obtained in the present study are clinically important and are worth validation in a larger, multicentric study enrolling more patients.

## 5. Conclusions

URS was significantly associated with worse OS and PFS in the subgroup of patients with non-papillary and ≥pT3 UTUC. Given the possible adverse oncological impact of diagnostic URS and the inaccuracy of image-based, clinical T staging, it may be safer to avoid diagnostic URS in patients with high-risk UTUC features unless URS is necessary to make a diagnosis of UTUC. Importantly, such adverse prognostic effects of URS were not observed in patients with papillary and ≤pT2 UTUC, corroborating the oncological safety of URS in patients with low-risk UTUC who can be managed conservatively using kidney-sparing therapy.

## Figures and Tables

**Figure 1 cancers-14-03962-f001:**
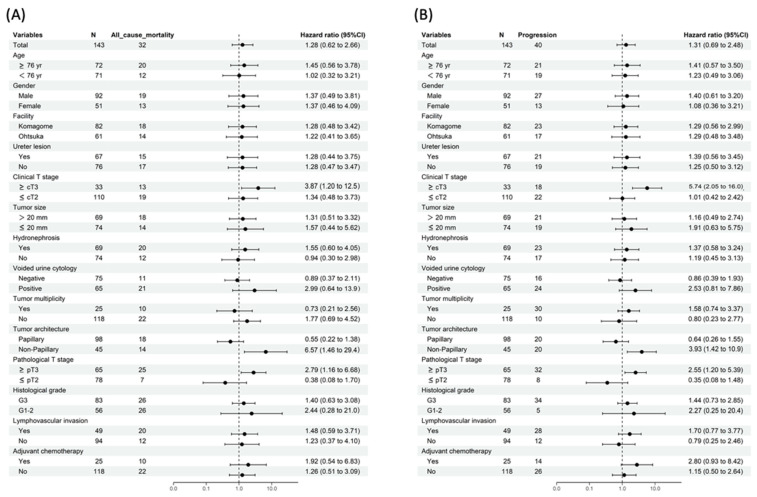
Subgroup analyses of the prognostic effects of ureterorenoscopy on overall survival (**A**) and progression-free survival (**B**).

**Figure 2 cancers-14-03962-f002:**
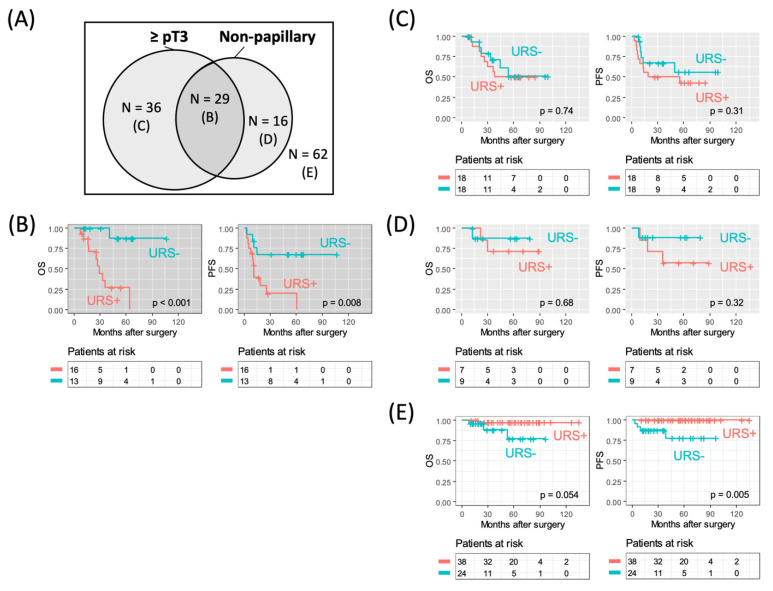
Demographics according to tumor architecture and pathological tumor stage (**A**) and Kaplan–Meier curves for overall survival (OS) and progression-free survival (PFS) according to whether or not a ureterorenoscopy (URS) was performed before radical nephroureterectomy in subgroups of patients with non-papillary upper tract urothelial carcinoma and ≥pT3 (**B**) disease, papillary and ≥pT3 (**C**) disease, non-papillary and ≤pT2 disease (**D**), and papillary and ≤pT2 disease (**E**). B to E in Panel (**A**) correspond to Panels (**B**–**E**), respectively.

**Table 1 cancers-14-03962-t001:** Demographics of 143 upper tract urothelial carcinoma patients according to whether or not ureterorenoscopy was performed prior to radical nephroureterectomy.

Variables	N (%)			*p* Value
	Total	URS+	URS−	
Total	143 (100)	79 (100)	64 (100)	
Age *	76 (69–79)	76 (70–79)	74 (68–79.75)	0.56
Gender				0.036
Male	92 (64.3)	57 (72.2)	35 (54.7)	
Female	51 (35.7)	22 (27.8)	29 (45.3)	
ECOG-PS				0.41
0	137 (95.8)	77 (97.5)	60 (93.8)	
1	6 (4.2)	2 (2.5)	4 (6.3)	
Body mass index *	23.1 (21.2–25)	22.9 (20.5–24.6)	23.3 (22–26.1)	0.052
Smoking history, pack year *	20 (0–47.7)	21 (0–43.8)	15.5 (0–50.8)	0.94
History of bladder cancer				0.49
Concurrent	16 (11.2)	7 (8.9)	9 (14.1)	
Previous	7(4.9)	5 (6.3)	2 (3.1)	
Never	120 (83.9)	67 (84.8)	53 (82.8)	
Facility				0.498
Komagome	82	43 (54)	39 (61)	
Ohtsuka	61	36 (46)	25 (39)	
Tumor location				0.61
Renal pelvis	76 (53.1)	39 (49.4)	37 (57.8)	
Ureter	61 (42.7)	36 (45.6)	25 (39.1)	
Both	6 (4.2)	4 (5.1)	2 (3.1)	
Clinical T stage				0.046
≤cT2	110 (66.9)	66 (83.5)	44 (68.7)	
≥cT3	33 (23.1)	13 (16.5)	20 (31.3)	
Clinical N stage				0.038
N0	139 (97.2)	79 (100)	60 (93.8)	
N1	4 (2.8)	0 (0)	4 (6.2)	
Tumor size (mm) *	18 (12–25)	15 (10–22)	22 (15–30)	<0.001
Hydronephrosis				
Yes	69 (48.3)	40 (50.6)	29 (45.3)	0.61
No	74 (51.7)	39 (44.4)	35 (54.7)	
Voided urine cytology				1
Positive	65 (45.5)	35 (44.3)	30 (46.9)	
Negative	76 (53.1)	42 (53.2)	34 (53.1)	
NA	2 (1.4)	2 (2.5)	0 (0)	
URS biopsy				<0.001
Yes	61 (42.7)	61 (77.2)	0 (0)	
No	82 (57.3)	28 (22.8)	64 (100)	
Surgical modality				0.736
Open	63 (44.1)	36 (45.6)	27 (42.2)	
Minimally invasive	80 (55.9)	43 (54.4)	37 (57.8)	
Tumor multiplicity				1
Yes	25 (17.5)	14 (17.7)	11 (17.2)	
No	118 (82.5)	65 (82.3)	53 (82.8)	
Tumor architecture				0.59
Papillary	98 (68.5)	56 (80.9)	42 (65.6)	
Non-papillary	45 (31.5)	23 (29.1)	22 (34.4)	
Pathological T stage				0.30
pTis/a/1	63 (44.1)	38 (48.1)	25 (39.1)	
pT2	15 (10.5)	7 (8.9)	8 (12.5)	
pT3	61 (42.7)	32 (40.5)	29 (45.3)	
pT4	4 (2.8)	2 (2.5)	2 (3.1)	
Pathological N stage				0.41
pN0	40 (28.0)	19 (24.1)	21 (32.8)	
pN1	6 (4.2)	2 (2.5)	4 (6.3)	
pNx	97 (67.8)	58 (73.4)	39 (60.9)	
Histological grade				0.30
G1	8 (5.6)	4 (5.1)	4 (6.3)	
G2	48 (33.6)	30 (38.0)	18 (28.1)	
G3	83 (58.0)	42 (53.2)	41 (64.1)	
NA	4 (2.8)	3 (3.8)	1 (1.6)	
Lymphovascular invasion				0.60
Yes	49 (34.3)	29 (36.7)	20 (31.3)	
No	94 (65.7)	50 (63.3)	44 (68.7)	
Positive surgical margin				0.83
Yes	6 (4.2)	4 (5.1)	2 (3.1)	
No	136 (95.1)	74 (93.7)	62 (96.9)	
Indeterminant	1 (0.7)	1 (1.3)	0 (0)	
Variant histology				0.25
Yes	13 (9.1)	5 (6.3)	8 (12.5)	
No	130 (90.9)	74 (93.7)	56 (87.5)	
Platinum-based adjuvant chemotherapy				0.51
Yes	25 (17.5)	12 (15.2)	13 (20.3)	
No	118 (82.5)	67 (84.8)	51 (79.7)	
All-cause mortality	32 (22.4)	21 (26.6)	11 (17.2)	
Progression	40 (28.0)	25 (31.6)	15 (23.4)	
Visceral or lymph node metastasis	35 (24.5)	23 (29.1)	12 (18.8)	
Local recurrence	5 (3.5)	2 (2.5)	3 (4.7)	

URS = ureterorenoscopy; ECOG-PS = Eastern Cooperative Oncology Group—performance status; NA = not available; * median (interquartile range).

**Table 2 cancers-14-03962-t002:** Variables associated with overall and progression-free survival in 143 upper tract urothelial carcinoma patients undergoing radical nephroureterectomy.

Variables	Overall Survival				Progression-Free Survival			
	Univariable		Multivariable		Univariable		Multivariable	
	HR (95% CI)	*p* Value	HR (95% CI)	*p* Value	HR (95% CI)	*p* Value	HR (95% CI)	*p* Value
Age	1.05 (1.00–1.10)	0.06			1.02 (0.98–1.05)	0.44		
Gender, male (vs. female)	0.73 (0.36–1.47)	0.38			1.14 (0.59–2.22)	0.69		
ECOG-PS, 1 (vs. 0)	4.23 (1.48–12.1)	0.007			2.32 (0.71–7.53)	0.16		
Body mass index	1.04 (0.96–1.13)	0.33			1.06 (0.99–1.14)	0.11		
Smoking history, pack year	1.00 (0.99–1.01)	0.79			1.00 (0.99–1.01)	0.81		
Facility, Komagome (vs. Ohtsuka)	1.04 (0.52–2.09)	0.92			0.99 (0.53–1.85)	0.97		
History of bladder cancer, yes (vs. no)	1.638 (0.73–3.63)	0.23			1.07 (0.47–2.41)	0.88		
Tumor location, ureter, or both (vs. renal pelvis)	1.12 (0.56–2.24)	0.75			1.44 (0.77–2.67)	0.25		
Clinical T stage, ≥cT3 (vs. ≤cT2)	2.98 (1.47–6.07)	0.003			3.97 (2.12–7.44)	<0.001		
Tumor size	1.02 (1.00–1.04)	0.11			1.02 (1.00–1.04)	0.14		
Hydronephrosis, yes (vs. no)	1.96 (0.96–4.00)	0.066			1.55 (0.83–2.91)	0.17		
Voided urine cytology, positive (vs. negative)	1.29 (0.64–2.60)	0.48			1.25 (0.67–2.35)	0.48		
URS, yes (vs. no)	1.28 (0.62–2.66)	0.50			1.31 (0.69–2.48)	0.41		
URS biopsy, yes (vs. no)	1.08 (0.54–2.16)	0.83			1.19 (0.64–2.22)	0.58		
Tumor multiplicity, yes (vs. no)	0.53 (0.25–1.12)	0.095			0.65 (0.32–1.32)	0.23		
Tumor architecture, non-papillary (vs. papillary)	2.01 (1.00–4.04)	0.051			2.61 (1.40–4.86)	0.003		
Pathological T stage, ≥pT3 (vs. ≤pT2)	5.28 (2.28–12.2)	<0.001	5.24 (2.26–12.1)	<0.001	6.49 (2.98–14.1)	<0.001	3.55 (1.47–8.55)	0.005
Pathological N stage, pN1 (vs. pN0)	2.14 (0.47–9.71)	0.32			2.81 (0.79–10.0)	0.11		
Histological grade, G3 (vs. G1–2)	3.45 (1.42–8.38)	0.006			5.63 (2.20–14.4)	<0.001		
Lymphovascular invasion, yes (vs. no)	4.11 (2.00–8.41)	<0.001	2.13 (0.95–4.76)	0.06	6.14 (3.11–12.1)	<0.001	3.10 (1.43–6.70)	0.004
Positive surgical margin, yes (vs. no)	3.68 (1.11–12.1)	0.033			5.78 (2.24–14.9)	<0.001		
Platinum-based adjuvant chemotherapy, yes (vs. no)	2.51 (1.19–5.32)	0.016			3.53 (1.82–6.85)	<0.001		
Variant histology, yes (vs. no)	2.45 (0.94–6.38)	0.065			3.43 (1.57–7.47)	0.002		

URS = ureterorenoscopy; ECOG-PS = Eastern Cooperative Oncology Group—performance status.

**Table 3 cancers-14-03962-t003:** Associations of image-based clinical T stage with pathological T stage.

Clinical T Stage	N		
	Total	≥pT3	≤pT2
Total	143	65	78
≥cT3	33	29	4
≤cT2	110	36	74

**Table 4 cancers-14-03962-t004:** Demographics of 29 upper tract urothelial carcinoma patients with ≥pT3 and non-papillary architecture according to whether or not ureterorenoscopy was performed prior to radical nephroureterectomy.

Variables	N (%)			*p* Value
	Total	URS+	URS−	
Total	29 (100)	16 (100)	13 (100)	
Age *	77 (69–79)	75 (66–79)	78 (70–83)	0.25
Gender				0.45
Male	20 (69)	10 (63)	10 (77)	
Female	9 (31)	6 (38)	3 (23)	
ECOG-PS				0.45
0	28 (96.6)	16 (100)	12 (92)	
1	1 (3.4)	0 (0)	1 (8)	
Body mass index *	23.8 (22.3–26.4)	23.9 (21.1–25.4)	23.5 (22.4–26.9)	0.68
Smoking history, pack year *	18 (0–49)	0 (0–33)	30 (5–57)	0.035
History of bladder cancer				0.19
Previous	1 (3)	1 (6)	0 (0)	
Concurrent	2 (7)	0 (0)	2 (15)	
No	26 (90)	15 (94)	11 (85)	
Facility				0.25
Komagome	18 (62)	8 (50)	10 (77)	
Ohtsuka	11 (38)	8 (50)	3 (23)	
Tumor location				0.84
Pelvis	13 (45)	6 (38)	7 (54)	
Ureter	15 (52)	9 (56)	6 (46)	
Both	1 (3)	1 (6)	0 (0)	
Clinical T stage				0.45
≤cT2	17 (59)	8 (50)	9 (69)	
≥cT3	12 (41)	8 (50)	4 (31)	
Clinical N stage				0.45
cN0	28 (97)	16 (100)	12 (92)	
cN1	1 (3)	0 (0)	1 (8)	
Tumor size *	15 (10–22)	14 (10–21)	17 (11–24)	0.36
Hydronephrosis				0.041
Yes	20 (69)	14 (88)	6 (46)	
No	9 (31)	2 (13)	7 (54)	
Voided urine cytology				1
Negative	12 (41)	8 (50)	4 (31)	
Positive	16 (55)	7 (44)	9 (69)	
NA	1 (3)	1 (6)	0 (0)	
URS biopsy				<0.001
Yes	12 (41)	12 (75)	0 (0)	
No	17 (59)	4 (25)	0 (0)	
Surgical modality				0.45
Open	12 (41)	8 (50)	4 (31)	
Minimally invasive	17 (59)	8 (50)	9 (69)	
Tumor multiplicity				1
Yes	3 (10)	2 (13)	1 (8)	
No	26 (90)	14 (88)	12 (92)	
Pathological T stage				1
pT3	25 (86)	14 (88)	11 (85)	
pT4	4 (14)	2 (13)	2 (15)	
Pathological N stage				0.57
pN0	6 (21)	4 (25)	2 (15)	
pN1	3 (10)	1 (6)	2 (15)	
pNx	20 (69)	11 (69)	9 (69)	
Histological grade				0.21
G2	2 (7)	2 (13)	0 (0)	
G3	27 (93)	14 (88)	13 (100)	
Lymphovascular invasion				0.27
Yes	17 (59)	11 (69)	6 (46)	
No	12 (41)	5 (31)	7 (54)	
Positive surgical margin				1
Yes	3 (10)	2 (13)	1 (8)	
No	25 (86)	13 (81)	12 (92)	
Indeterminant	1 (3)	1 (6)	0 (0)	
Variant histology				0.63
Yes	5 (17)	2 (13)	3 (23)	
No	24 (83)	14 (88)	10 (77)	
Platinum-based adjuvant chemotherapy				1
Yes	14 (48)	8 (50)	6 (46)	
No	15 (52)	8 (50)	7 (54)	
All-cause mortality	11 (38)	10 (63)	1 (8)	
Progression	16 (55)	12 (75)	4 (31)	
Visceral or lymph node metastasis	13 (45)	11 (69)	2 (15)	
Local recurrence	3 (10)	1 (6)	2 (15)	

URS = ureterorenoscopy; ECOG-PS = Eastern Cooperative Oncology Group—performance status; NA = not available; * Median (interquartile range).

**Table 5 cancers-14-03962-t005:** Variables associated with overall and progression-free survival in the subgroup of non-papillary architecture and ≥pT3 (n = 29).

Variables	Overall Survival		Progression-Free Survival			
	Univariable		Univariable		Multivariable	
	HR (95% CI)	*p* Value	HR (95% CI)	*p* Value	HR (95% CI)	*p* Value
Age	0.99 (0.92–1.06)	0.71	0.98 (0.93–1.03)	0.45		
Smoking history, pack year	0.98 (0.95–1.01)	0.21	1.00 (0.98–1.02)	0.89		
Tumor location, ureter or both (vs. renal pelvis)	0.94 (0.28–3.11)	0.91	0.74 (0.28–2.00)	0.56		
Clinical T stage, ≥cT3 (vs. ≤cT2)	1.43 (0.41–4.95)	0.57	2.25 (0.81–6.27)	0.12		
Tumor size	1.02 (0.94–1.11)	0.67	1.03 (0.96–1.1)	0.41		
Hydronephrosis, yes (vs. no)	0 (0–infinity)	1	2.54 (0.72–8.96)	0.15	1.32 (0.32–5.39)	0.70
Voided urine cytology, positive (vs. negative)	0.79 (0.22–2.86)	0.72	1.01 (0.36–2.87)	0.98		
URS, yes (vs. no)	15.8 (1.97–126)	0.009	4.30 (1.36–13.7)	0.013	3.83 (1.06–13.9)	0.041
URS biopsy, yes (vs. no)	5.23 (1.50–18.2)	0.009	2.04 (0.75–5.54)	0.16		
Pathological N stage, N1 (vs. N0)	1.55 (0.16–15.1)	0.71	1.66 (0.30–9.18)	0.56		
Histological grade, G3 (vs. G2)	0.36 (0.08–1.73)	0.20	0.49 (0.11–2.23)	0.36		
Lymphovascular invasion, yes (vs. no)	2.06 (0.60–7.09)	0.25	2.05 (0.70–5.97)	0.19		
Positive surgical margin, yes or indeterminate (vs. no)	2.86 (0.58–14.0)	0.19	5.36 (1.36–21.2)	0.016		
Platinum-based adjuvant chemotherapy, yes (vs. no)	1.87 (0.56–6.21)	0.31	1.83 (0.68–4.99)	0.23		

URS = ureterorenoscopy.

## Data Availability

Data sharing is not applicable to this article.

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
