# Peer review of "Adverse Prognostic Impact of Diagnostic Ureterorenoscopy in a Subset of Patients with High-Risk Upper Tract Urothelial Carcinoma Treated with Radical Nephroureterectomy"

_cancers, 2022, doi:10.3390/cancers14163962_

Round 1

Reviewer 1 Report

 The authors evaluate the impact of diagnostic ureteroscopy on progression free and overall survival following nephroureterectomy.  When evaluating the entire cohort diagnostic ureteroscopy did not impact progression free survival or overall survival.  Subgroup analysis suggested that patients with >pT3 and non papillary disease had decreased survival with diagnostic ureteroscopy prior to nephroureterectomy.   The presented results must be viewed with caution as unplanned subgroup analysis was required to note a statistical significance in survival.  A pre analysis hypothesis explaining the need for subgroup analysis would have strengthened the finds.  

The author's best point is that diagnostic ureteroscopy should be limited to equivocal cases or where a biopsy will change management.

1. Would remove N+ patients from analysis as these patients already fit the definition of progression utilized in the survival analysis.

2. Was timing and manner of follow up similar between groups?

3. It would be helpful to the reader if the univariate and multivariate analysis results were presented in a table.

Reviewer 2 Report

Dear Authors,

thank you for submitting your work.

As you mention in the first sentence of your abstract that you hypothesize that diagnostic ureterorenoscopy (URS) may adversely affect prognosis in a subset of patients with high-risk upper tract urothelial carcinoma (UTUC) undergoing radical nephroureterectomy (RNU), I would recommend focusing on this cohort - meaning only including patients with high-risk UTUC and then compare the group that had URS vs the group that had no URS. It would be much more informative and find its value in future pooled analyses.

You mention in your introduction that URS is known to increase intravesical recurrence after RNU. Hence, I would recommend including the endpoint 'intravesical recurrence-free survival' in your study. It would also be interesting to see which patients got intravesical chemotherapy after URS (if applicable).

As you compiled data from two centers only, it would be more accurate to define your study as being a bicentric retrospective cohort study, instead of "multicentric".

Please ensure that you include all UTUC prognostic factors for high-risk UTUC in Table 1. Of note, I could not find Table 2 and Table 5 embedded in the manuscript.

Further, please revise Figure 2 - probably taking out 2A and reformatting it. Of note, I could not find a Figure 1 in the manuscript.

Before resubmitting, please also make sure that your manuscript is in accordance with the STROBE checklist for cohort studies (https://www.equator-network.org/reporting-guidelines/strobe/).

Thanks again for your valuable work and for submitting it to this journal!

Round 2

Reviewer 1 Report

The authors have adequately addressed by suggestions and queries.

Author Response

We greatly appreciate the Reviewer for giving us constructive suggestions! 

Reviewer 2 Report

Thank you for re-submitting your work!

I have two more comments:

Please indicate the correct number of studied individuals that you analyzed in Figure 1. I understand that your total cohort is 143 patients with upper tract urothelial carcinoma undergoing radical nephroureterectomy. However, if it is a subgroup analysis, there should be fewer patients.

Please reformat Figure 2. The survival graphics should have the same proportions.

Thanks.

Author Response

We greatly appreciate the Reviewers for taking their precious time to review our paper again. We addressed the points raised by the Reviewer 2 as follows.

Thank you for re-submitting your work! 

I have two more comments:

Please indicate the correct number of studied individuals that you analyzed in Figure 1. I understand that your total cohort is 143 patients with upper tract urothelial carcinoma undergoing radical nephroureterectomy. However, if it is a subgroup analysis, there should be fewer patients.

We have indicated the number of patients for each subgroup (age, gender, …) in the column “N” next to the column “variables” in Figures 1A and 1B.

Please reformat Figure 2. The survival graphics should have the same proportions.

We revised Figure 2 in accordance with the Reviewer’s suggestion.

Thanks.
